# Population genomics of intrapatient HIV-1 evolution

**Fabio Zanini[1], Johanna Brodin[2], Lina Thebo[2], Christa Lanz[1], Göran Bratt[3], Jan Albert[2,4]\*, Richard A Neher[1]\***

[1]Evolutionary Dynamics and Biophysics, Max Planck Institute for Developmental Biology, Tübingen, Germany; [2]Department of Microbiology, Tumor and Cell Biology, Karolinska Institute, Stockholm, Sweden; [3]Department of Clinical Science and Education, Stockholm South General Hospital, Stockholm, Sweden; [4]Department of Clinical Microbiology, Karolinska University Hospital, Stockholm, Sweden

**Abstract** Many microbial populations rapidly adapt to changing environments with multiple variants competing for survival. To quantify such complex evolutionary dynamics in vivo, time resolved and genome wide data including rare variants are essential. We performed whole-genome deep sequencing of HIV-1 populations in 9 untreated patients, with 6-12 longitudinal samples per patient spanning 5-8 years of infection. The data can be accessed and explored via an interactive web application. We show that patterns of minor diversity are reproducible between patients and mirror global HIV-1 diversity, suggesting a universal landscape of fitness costs that control diversity. Reversions towards the ancestral HIV-1 sequence are observed throughout infection and account for almost one third of all sequence changes. Reversion rates depend strongly on conservation. Frequent recombination limits linkage disequilibrium to about 100bp in most of the genome, but strong hitch-hiking due to short range linkage limits diversity.

**\*For correspondence:** Jan. Albert@ki.se (JA); richard.neher@ tuebingen.mpg.de (RAN)

## Introduction

The human immunodeficiency virus 1 (HIV-1) is a paradigmatic example of a rapidly adapting population characterized by high diversity, strong selection, and recombination. HIV-1 has originated from multiple zoonotic transmissions from apes in the early part of the 20th century (*Sharp and Hahn, 2011*), one of which gave rise to the worldwide pandemic. This lineage is called group M and has diversified into different subtypes at a rate of about 1 in 1000 substitutions per year (*Lemey et al., 2005*; *Li et al., 2015*). Tens of thousand of HIV-1 group M variants are available in the Los Alamos National Laboratories HIV database (LANL) (*Foley et al., 2013*).

The evolution of HIV-1 ultimately takes place within infected individuals and can be observed directly in longitudinal samples of virus populations from the same individual. The detailed knowledge of HIV-1 biology paired with historical samples makes HIV-1 an ideal system to study general features of evolution at high mutation rates and strong selection that are otherwise only accessible in evolution experiments (*Elena and Lenski, 2003*; *Miralles et al., 1999*).

During the first few months of an HIV-1 infection, the viral population typically acquires several mutations that mediate escape from cytotoxic T-cells (CTL). The properties and dynamics of these mutations and their effect on epitope presentation and virus control have been extensively studied (*Bernardin et al., 2005*; *Goonetilleke et al., 2009*; *Jones et al., 2004*; *Kearney et al., 2009*; *Liu et al., 2011*). Such escape variants often compromise virus replication and have been shown to revert upon transmission of the virus to a human leukocyte antigen (HLA) mismatched host (*Brockman et al., 2010*; *Leslie et al., 2004*; *Li et al., 2007*). However, the quantitative contribution

**eLife digest** HIV was transmitted from apes to humans multiple times before the virus began to spread among humans some time in the early 20th century. The virus is now found around the globe and has evolved into several different subtypes. Also, HIV continues to evolve inside each infected individual so that different variations of the virus are often present in the same person at any given time.

When a person initially becomes infected, immune cells called killer T-cells seek out and destroy infected cells. However, HIV quickly acquires genetic mutations that allow it to escape the immune system and multiply in the body. There is a cost to this evasion strategy: the mutant forms of the virus often don't multiply as rapidly as the variant that first infected the individual. This may explain why the virus often mutates back to its previous form in newly infected people. But many questions remain about how these pressures influence the evolution of the virus in individuals and in populations over time.

Now, Zanini et al. have sequenced the genomes of HIV samples collected from nine people with HIV in Sweden over the course of five to eight years. These individuals were diagnosed between 1990 and 2003 – when HIV medications were not as widely used as they are now – and did not receive any treatment during the course of the study. Therefore, the study provides a rare opportunity to look at how HIV evolves in the absence of drugs that target the virus.

Zanini et al. sequenced the entire genetic code of each form of the virus identified in the samples. The data show that HIV follows predictable patterns of evolution within individuals as well as across human populations. Mutations happen frequently all over the HIV genome. However, the mutant viruses often revert to a common or "optimal" form of the virus throughout the course of infection. This suggests that there is a tradeoff between the benefits of acquiring new mutations and maintaining a set of traits that have enabled the virus to spread so successfully in humans.

To make it easier for other researchers to explore the data, Zanini et al. created a web application that allows others to access and create visual representations of viral evolution.

Together these findings suggest that it will be possible to achieve a fuller understanding of RNA virus evolution that integrates the molecular biology of the virus and the immune response of the host with the evolutionary changes.

of escape and reversion to HIV-1 evolution and the degree to which costly mutations are compensated by additional mutations is less well understood (*Lythgoe and Fraser, 2012*; *Schneidewind et al., 2009*). Similar to CTL escape mutations, drug resistance mutations can spread through the viral population within weeks and longitudinal sequence data has been used to study emergence and fitness cost of resistance mutations (*Hedskog et al., 2010*; *Little et al., 2008*; *Paredes et al., 2009*).

Longitudinal sequence data has also been used to track evolution driven by the humoral immune response against HIV-1 which occurs throughout infection and results in high rates of evolution in the variable loops of HIV-1 surface proteins (*Bar et al., 2012*; *Richman et al., 2003*; *Shankarappa et al., 1999*). In a pioneering study, *Shankarappa et al. (1999)* characterized HIV-1 evolution of parts of the gp120 envelope protein over approximately 6–12 years in 11 patients, demonstrating consistent patterns of diversity and divergence. While this early study was limited to about 10 sequences per sample, next-generation sequencing technology today allows deep characterization of intrapatient HIV-1 variability including rare mutations (*Fischer et al., 2010*; *Hedskog et al., 2010*; *Henn et al., 2012*; *Tsibris et al., 2009*).

The earlier studies discussed above have either focused on early infection or particular regions of the HIV-1 genome. However, most virus evolution happens during chronic infection simultaneously at many locations in the genome. To develop a comprehensive and quantitative understanding of the evolution and diversification of HIV-1 populations, we generated a whole-genome, deep-sequencing data set covering nine patients over 5–8 years with 6–12 time points per patient. Importantly, the data set covers the entire genome such that no substitution is missed and includes early samples defining the initial population. To our knowledge, this is the only whole-genome deep sequencing data set with long follow-up of multiple patients. We provide interactive and intuitive

web-based access to the data set and hope it will become a resource for many future studies, like the data set by *Shankarappa et al. (1999)* has been in the last years.

Below, we first describe the methodology we developed to sequence the entire HIV-1 genome at great depth. We then analyze the intrapatient evolution of HIV-1 and show that the minor variants in the virus population explore sequence space in predictable fashion at the single site level. At the same time, we observe a strong tendency for reversion towards the global HIV-1 consensus that is not limited to early infection but occurs at an approximately constant rate throughout chronic infection. Reversion is more frequent at sites that are more conserved at the global level, suggesting a direct relationship between intrapatient fitness cost and global conservation. Together with reproducible patterns of intrapatient variation, this link explains why HIV-1 fitness landscapes can be inferred from cross-sectional data (*Dahirel et al., 2011*; *Ferguson et al., 2013*; *Mann et al., 2014*). We further find frequent recombination, which allows the viral population to evolve independently in different regions of the genome. Nevertheless, recombination is not frequent enough to decouple mutations closer than 100 base pairs and we observe signatures of hitch-hiking at short distances (*Maynard Smith and Haigh, 1974*).

## Results

The study included nine HIV-1-infected patients who were diagnosed in Sweden between 1990 and 2003. Data from two additional patients (p4 and p7) were excluded from analysis because of suspected superinfection and failure to amplify early samples with low virus levels, respectively. The patients were selected to have a relatively well-defined time of infection and to have been treatment-naive for a minimum of 5 years. Patients diagnosed in recent years rarely fulfill these inclusion criteria because therapy is almost universally recommended, but this was not the case when the study patients were diagnosed. Basic characteristics of the patients are presented in *Table 1*. Additional data on the patients are presented in *Supplementary file 1* and *Figure 1—figure supplement 1*. All patients progressed to moderate-to-severe immunosuppression during the study period. The final CD4 count within the study period for patients 1–3, 5–6 and 8–11 were 340, 369, 140, 228, 287, 378, 158, 200 and 251, respectively. Of the nine patients, eight were males, seven were men who have sex with men (MSM), two were heterosexually infected (HET), and eight were infected in Sweden. The method to estimate the date of infection (EDI) is described in Materials and

**Table 1.** Summary of patient characteristics. Sample times from estimated date of infection.

| Patient | Gender | Transmission route | Subtype | Age* [years] | Fiebig stage* | BED* [ODn] | No. of samples | First sample [days] | Last sample [years] | HLA type A | B | C |
|---|---|---|---|---|---|---|---|---|---|---|---|---|
| p1 | F | HET | 01_AE | 37 | VI | 0.42 | 12 | 122 | 8.2 | 02/02 | 08/15 | 03/06 |
| p2 | M | MSM | B | 32 | V | 0.17 | 6 | 74 | 5.5 | 01/24 | 08/39 | 07/12 |
| p3 | M | MSM | B | 52 | VI | 0.90 | 10 | 146 | 8.4 | 02/11 | 15/44 | 03/16 |
| p5 | M | MSM | B | 38 | VI | 0.15 | 7 | 134 | 5.9 | 03/33 | 14/58 | 03/08 |
| p6 | M | HET | C | 31 | IV | 0.29 | 7 | 62 | 7.0 | 02/02 | 44/51 | 05/16 |
| p8 | M | MSM | B | 35 | V | 0.16 | 7 | 87 | 6.0 | 03/32 | 07/40 | 02/07 |
| p9 | M | MSM | B | 32 | VI | 0.28 | 8 | 106 | 8.1 | 25/32 | 07/44 | 04/07 |
| p10 | M | MSM | B | 34 | II | 0.10 | 9 | 33 | 6.2 | 32/32 | 44/50 | 06/16 |
| p11 | M | MSM | B | 53 | VI | 1.22 | 7 | 209 | 5.6 | 02/32 | 39/44 | 05/12 |

*, at base line; MSM, men who have sex with men; HET, heterosexual; MSM, men who have sex with men.

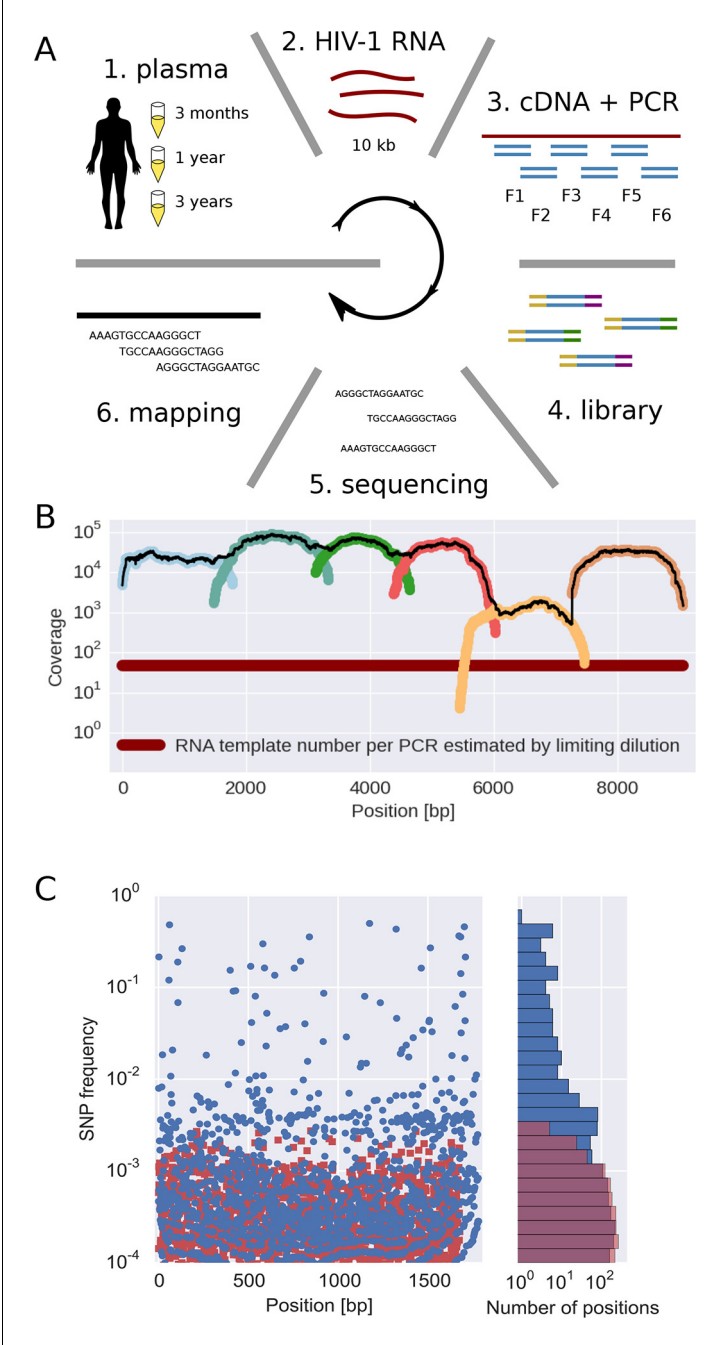

**Figure 1.** Sequencing, coverage, and error rates. (**A**) Schematics of the sample preparation protocol, see text and Materials and methods for details. (**B**) Read coverage for a representative sample. Coverage of separate PCR amplicons is shown in different hues, the black line is the total coverage. The coverage of PCR fragment F5 is lower than the other amplicons, but it is still larger than number of input HIV-1 RNA molecules; this situation is typical in our samples. (**C**) Each blue circle corresponds to a SNP frequency in amplicon F1 of a late sample of patient 11, while red squares are SNP frequencies in the sequence data generated from 10,000 copies of plasmid NL4-3. The histogram on the right shows the distribution of SNP frequencies in the patient sample and the control. Minor SNPs observed in reads generated from the plasmid, which represent PCR and sequencing errors, did not exceed 0.3%. SNP, single nucleotide polymorphism; RT-PCR, reverse transcriptase polychromase chain reaction.

The following figure supplements are available for figure 1:

*Figure 1. continued on next page*

*Figure 1. Continued*

**Figure supplement 1.** Viral load and CD4[+] cell counts for all nine patients.

**Figure supplement 2.** Tree of minor reads covering p17 from all samples colored by patient ID.

methods. We retrieved longitudinal biobank plasma samples covering the time period from early infection (time since EDI 33–209 days) until end of study follow-up, which usually coincided with start of antiretroviral therapy (time since EDI 5.5–8.3 years). The number of samples per patient ranged between 6 and 12 and, in total, we investigated 73 samples. The median plasma HIV-1 RNA level was 12,000 copies/ml for the samples with available data on RNA levels. Some samples had low RNA levels, which partly is explained by the study design that required that the patients were treatment-naive for at least 5 years. For detailed information about each individual sample see *Supplementary file 1*. For a phylogenetic tree of minor variants from all patients see *Figure 1—figure supplement 2*.

## HIV-1 whole-genome deep sequencing

The basic steps of our amplification and sequencing pipeline are illustrated in *Figure 1* and explained in detail in Materials and methods. Briefly, viral RNA was extracted from 400 $\mu$l patient plasma and used for one-step RT-PCR amplification with six overlapping primer sets that span almost the complete HIV-1 genome, similar to the strategy developed by *Gall et al. (2012)*. Sequencing libraries were made starting from 0.1–1.5 ng of DNA with a stringent size selection for long inserts (>400 bp). Sequencing was performed on the Illumina MiSeq platform and sequence reads were quality filtered and assembled using an in-house data processing pipeline. In total, approximately 100 million reads passed the quality filtering. The coverage varied considerably between samples and amplicons, but was mostly of the order of several thousands or more, see *Figure 1B*.

Importantly, sequencing depth is determined not only by coverage but also by template availability and sequencing errors (*Iyer et al., 2015*). We performed a number of control experiments to quantify templates and assess the accuracy of estimates of frequencies of single nucleotide polymorphism (SNP). The results are summarized in the following section and described in detail in Materials and methods.

We quantified the number of HIV-1 genomes that contributed to each sequencing library by PCR-limiting dilution (median: 120 quartiles: 50–500). Hence template availability, rather than coverage, determined the sequencing depth, see *Figure 1B*. Comparison with routine plasma HIV-1 RNA level measurements performed at time of sampling showed that the median template complementary DNA (cDNA) recovery efficiency was 30%.

We estimated the error rate of the PCR and sequencing pipeline by amplifying and sequencing a plasmid clone. *Figure 1C* compares the SNP frequencies observed in a clone to those observed in a patient sample. After quality filtering, PCR and sequencing errors never exceeded 0.3% of reads covering a particular position. To detect and control for variation in PCR efficiency among fragments and skewed amplification of different variants, we compared frequencies of variants in overlaps between the six amplicons. A SNP in the overlap is amplified and sequenced twice independently and the concordance of the two measurements of variant frequencies was used to estimate the fragment specific depth, which is limited both by template input and PCR efficiency, see Materials and methods. Frequency estimates were often reproducible to within a few percent. Sometimes, however, variant recovery was poor (mostly in fragment 5) and frequency estimates less accurate. Those fragments are flagged on the website and can be excluded from analyses requiring large depth.

We minimized PCR recombination by reducing the number of PCR cycles and optimizing the reaction protocol (see Materials and methods, *Di Giallonardo et al. (2013)*; *Mild et al. (2011)*). Control experiments using mixtures of two cultured virus populations showed that less than 10% of reads had experienced RT-PCR recombination.

Taken together, our control experiments show that depending on the sample and fragment, we could estimate frequencies of SNPs down to 1% accuracy (corresponding to several hundred effective templates). In some cases, however, the template number was low or template recovery poor

such that only presence or absence of a variant could be called. Furthermore, SNPs remained linked through cDNA synthesis, PCR and sequencing.

## Website

Deep sequencing data sets like the one presented here require substantial filtering, mapping, and processing before they can be used to address specific biological questions. Hence depositing the raw data (while important) is of limited use for follow-up analyses. To facilitate reuse of our data, we developed a web application to explore the data set interactively and visualize the patterns of HIV-1 evolution in the study patients. In addition, the website provides easy access to processed data. These include phylogenetic trees, viral loads and CD4+ T cell counts, consensus sequences, major and minor haplotypes in different regions of the genome, frequencies of single nucleotide polymorphisms, and clean sequencing reads. The application allows browsing the data either by patient or by genomic region and provides composite interactive plots to explore how the virus population has changed over time. We hope that this web application, available at hiv.tuebingen.mpg.de, will encourage others to further analyze these data.

## Consistent evolution across the entire genome

In most patients, the virus populations were initially homogeneous and diversified over the years, as expected for an infection with a single or small number of similar founder viruses (*Keele et al., 2008*; *Salazar-Gonzalez et al., 2009*). In two patients, p3 and p10, the first sample displayed diversity consistent with the transmission of several variants from the same donor, see *Figure 3—figure supplement 1*. For each of the nine patients, we reconstructed the HIV-1 genome sequence of the first sample by an iterative mapping procedure described in Materials and methods. We use this initial consensus sequence to approximate the sequence of the transmitted founder virus(es). Our first sample is estimated to be between 1 and 7 months into infection and a few mutations (likely CTL escape mutations) had probably spread through the viral population by that time. Thus, the initial consensus sequence probably will differ slightly from the true founder virus. The number of differences, however, will be small compared to the sequence divergence in the 5 or more years of follow-up such that this initial consensus sequence is a useful approximation of the founder virus(es).

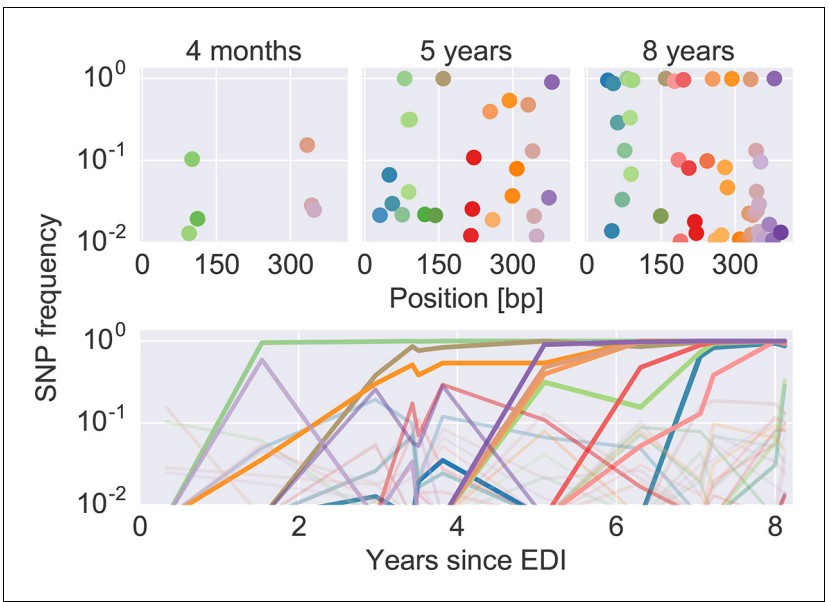

**Figure 2.** The dynamics of SNP frequencies. The upper panels show single nucleotide polymorphism frequencies along p17 at three time points in patient p1. The lower panel shows the trajectories of SNPs through time. Color corresponds to position in the sequence. Trajectory that reach high frequencies are shown with thicker and more opaque lines. Analogous data is available for all patients for most of the HIV-1 genome. EDI, estimated date of infection; SNP, single nucleotide polymorphism.

*Figure 2* shows an example of the dynamics of frequencies of SNP relative to the founder sequence over time, where each dot (top) or line (bottom) represents the frequency of a nucleotide different from the founder sequence. Interactive versions of this graph are available for the entire genome of all patients at hiv.tuebingen.mpg.de.

To measure the rate at which the virus population accumulates mutations, we calculated the average distance of each sample from the approximate founder sequence in 300 bp windows. Regressing this distance against time yields the rate of divergence in different regions of the genome, see *Figure 3A*. As expected, some regions such as the variable loops in gp120 and nef evolve faster, while enzymes – protease (PR), reverse transcriptase (RT) – and the rev response element (RRE) evolve more slowly. The rate of divergence varies by about a factor of 10 along the genome, but is consistent with typically about 1.5-fold differences across patients (standard deviation of log2(fold change) 0.6 ± 0.2). The overall pattern of the rate of mutation accumulation agrees with a recent map of HIV genome-wide variation from a population perspective (*Li et al., 2015*) and correlates well with entropy in a large HIV-1 group M alignment (Spearman's p = 0.7 after the same smoothing).

## Minor variants reproducibly explore global HIV-1 diversity

Having found that coarse patterns of divergence are comparable among patients, we asked whether intrapatient diversity at individual sites in the viral genome follows general and predictable patterns. To this end, we compared diversity at each position to the diversity observed in HIV-1 group M (see Materials and methods).

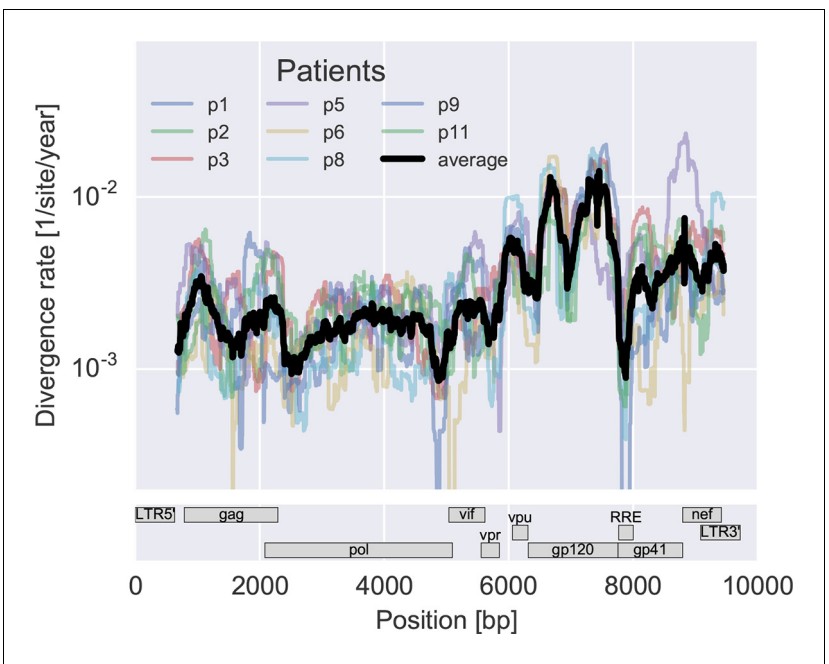

**Figure 3.** Consistent evolution across the viral genome. The figure shows the rate of sequence divergence averaged in a sliding window of length 300 bp for individual study participants (in color) and averaged over all (black). Rapidly evolving (V loops in gp120) and conserved (RRE) regions are readily apparent. The divergence rates are determined by linear regression of the distance from the putative founder sequence against time since EDI. This distance includes contributions of minor variants. All positions are given in HxB2 numbering. The corresponding figure for amino acid evolution is provided as *Figure 3—figure supplement 2*.

The following figure supplements are available for figure 3:

**Figure supplement 1.** Diversity over time for all six PCR fragments and all patients.

**Figure supplement 2.** Divergence rate like *Figure 3* but for amino acids rather than nucleotides.

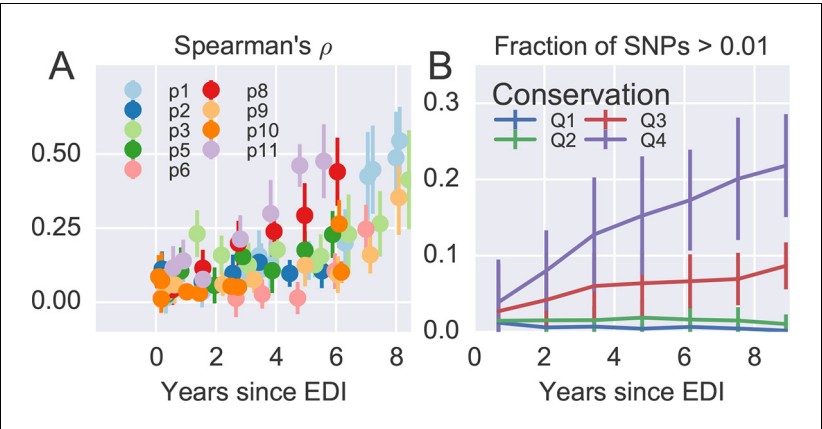

**Figure 4.** Within patient variation mirrors global variation. (**A**) Intrapatient variation at individual sites is correlated with diversity at homologous positions in an alignment of sequences representative of HIV-1 group M. This correlation increases reproducibly throughout the infection. Error bars show standard deviations over genomic regions. (**B**) Similarly, the fraction of sites with minor variants above 1% increases over time at the least constrained positions (quartiles Q3 and Q4), while few sites in the most conserved quartiles (Q1 and Q2) are polymorphic. *Figure 4—figure supplements 1* and 2 show the corresponding results for amino acid rather than nucleotide comparisons and patient–patient correlations of diversity, respectively. EDI, estimated date of infection.

The following source data and figure supplements are available for figure 4:

**Source data 1.** Tab-delimited files with plotted data.

**Figure supplement 1—Source data 1.** Tab-delimited files with plotted data.

**Figure supplement 1.** Within–patient variation mirrors global variation at the amino acid level, similar to what we observed at the nucleotide level.

**Figure supplement 2.** Correlation in nucleotide variation between study participants.

*Figure 4A* shows the rank correlation between the site-by-site diversity in each patient and a global collection of HIV-1 sequences, both measured by Shannon entropy (see Materials and methods). In all cases, correlation with cross-sectional diversity was initially low, as expected for largely homogeneous populations. As diversity increases within patients, it tends to accumulate at positions that are not conserved, resulting in a rank correlation of about 0.4–0.5 after about 8 years. These correlations are individually significant and reproducible among different genomic regions (error bars in *Figure 4*).

*Figure 4B* offers an alternative perspective on the exploration of sequence space by the HIV-1 populations. We classified nucleotide positions in the genome into four categories ranging from highly conserved positions to less conserved positions within group M (Q1–Q4) using the same alignment as above. Next we asked what fraction of sites within these categories show intrapatient variation at a level of at least 1%. For the least conserved positions, this fraction increased steadily to about 20% after 8 years, while less than 1% of the most conserved sites shows variation above the 1% level. This latter fraction rapidly saturates and does not increase over time. Since variant amplification, sequencing and variant calling does not use any information on cross-sectional conservations, the absence of variation above 1% at conserved sites is further evidence that our amplification and sequencing pipeline does not generate spurious variation. Other thresholds yield similar results.

Taken together, the observations in *Figure 4* show that variation is not limited by the mutational input, but that HIV-1 populations accumulate diversity wherever mutations do not severely compromise virus replication. At the single nucleotide level, the spectrum of mutational possibilities is explored reproducibly and the level of within-host diversity is predicted by time since infection and cross-sectional diversity. Conserved positions are typically monomorphic even in deeply sequenced samples with high RNA template input.

## The majority of nonsynonymous substitutions are positively selected

In addition to the reproducible patterns of minor diversity, virus evolution is characterized by adaptations that are specific to the host. Immune selection results in escape mutations that rapidly spread through the population (*Richman et al., 2003*; *Walker and McMichael, 2012*). Such mutations tend to be nonsynonymous, i.e., change the viral proteins, while evolution at synonymous sites is expected to be conservative. Nevertheless, synonymous mutations can be affected by 'selective sweeps' of linked nonsynonymous mutations (*Maynard Smith and Haigh, 1974*).

To quantify the degree at which the evolutionary dynamics of HIV-1 is dominated by selective sweeps, we calculated divergence and diversity separately at nonsynonymous and synonymous sites in different parts of the HIV-1 genome. *Figure 5A* compares nonsynonymous (solid lines) and synonymous divergence (dashed lines) in different regions of the genome. In agreement with the results presented in *Figure 3*, the observed rate of evolution at nonsynonymous sites differed substantially between genomic regions, with env being the fastest and pol the slowest. Divergence at synonymous sites, however, varied very little between different genomic regions indicating random accumulation of synonymous mutations (rather than positive selection).

*Figure 5B* shows the corresponding plot for diversity, i.e., the distances between sequences from the same sample. Diversity at nonsynonymous sites (solid lines) saturates after about 2–4 years, suggesting that nonsynonymous SNPs either stay at low frequency because they are deleterious or rapidly increase in frequency and fix without contributing much to diversity. Synonymous diversity increases steadily in the 5' part of the genome (structural and enzymes), while it saturates in the 3' half of the genome after a few years – the exact opposite of nonsynonymous divergence.

Indeed, we observe a strong anti-correlation between synonymous diversity and nonsynonymous divergence, which is further quantified in *Figure 5C*. This suggests that frequent non-synonymous substitutions limit synonymous diversity because they drive linked synonymous mutations to fixation or to extinction (*Maynard Smith and Haigh, 1974*). We will quantify linkage and recombination below, but the differences in diversity accumulation already suggest that linkage is restricted to short distances.

The contrasting behavior of synonymous and nonsynonymous SNPs is also seen in the SNP frequency spectrum – the histogram of SNP abundance – shown in *Figure 5D*. While the spectra agree for frequencies below 20%, synonymous mutations are strongly underrepresented at higher frequencies (Fisher's exact test at frequency 0.5, p-value $<10^{-10}$). This corroborates the interpretation that, due to substantial recombination, sweeping nonsynonymous mutations only occasionally 'drag' adjacent synonymous mutations to fixation. Synonymous mutations rarely rise in frequency because of their own effect on fitness, since they usually have small or deleterious phenotypic effects and do not contribute directly in immune evasion (*Zanini and Neher, 2013*). The about five-fold excess of nonsynonymous over synonymous SNPs at high frequencies (see *Figure 5D*) shows that the majority of common nonsynonymous mutations spread due to positive selection.

Next, we sought to quantify what fraction of nonsynonymous divergence is driven by escape from cytotoxic T-lymphocytes (CTLs). Four-digit HLA types were determined for all patients and a set of putatively targeted HIV-1 epitopes were determined using the epitope binding prediction tool MHCi (tools.immuneepitope.org/mhci). We then asked whether we observed more nonsynonymous substitutions in epitopes predicted to be targeted than expected by chance (excluding the variable loops of gp120 and the external part of gp41, see Materials and methods). We found a significant enrichment by a factor 1.9 in the putatively targeted region (p-value $<3 \times 10^{-6}$), corresponding to 5.5 excess nonsynonymous substitutions, whereby the total number of nonsynonymous substitutions per patient is on average 43 (median, quartiles 36–64). The set of predicted epitopes will contain false positives and lack true epitopes, hence the actual number of CTL driven substitutions could be higher as for example suggested by *Allen et al. (2005)* who report that roughly half of non-envelope mutations are associated with CTL responses.

## Extensive reversion towards consensus

Many CTL escape mutations reduce the replicative capacity of the virus and it is known that such escape mutations often revert upon transmission to a host in which the corresponding epitope is not targeted (*Friedrich et al., 2004*; *Herbeck et al., 2006*; *Leslie et al., 2004*). The balance between escape and reversion results in association between specific escape mutations and the HLA types of

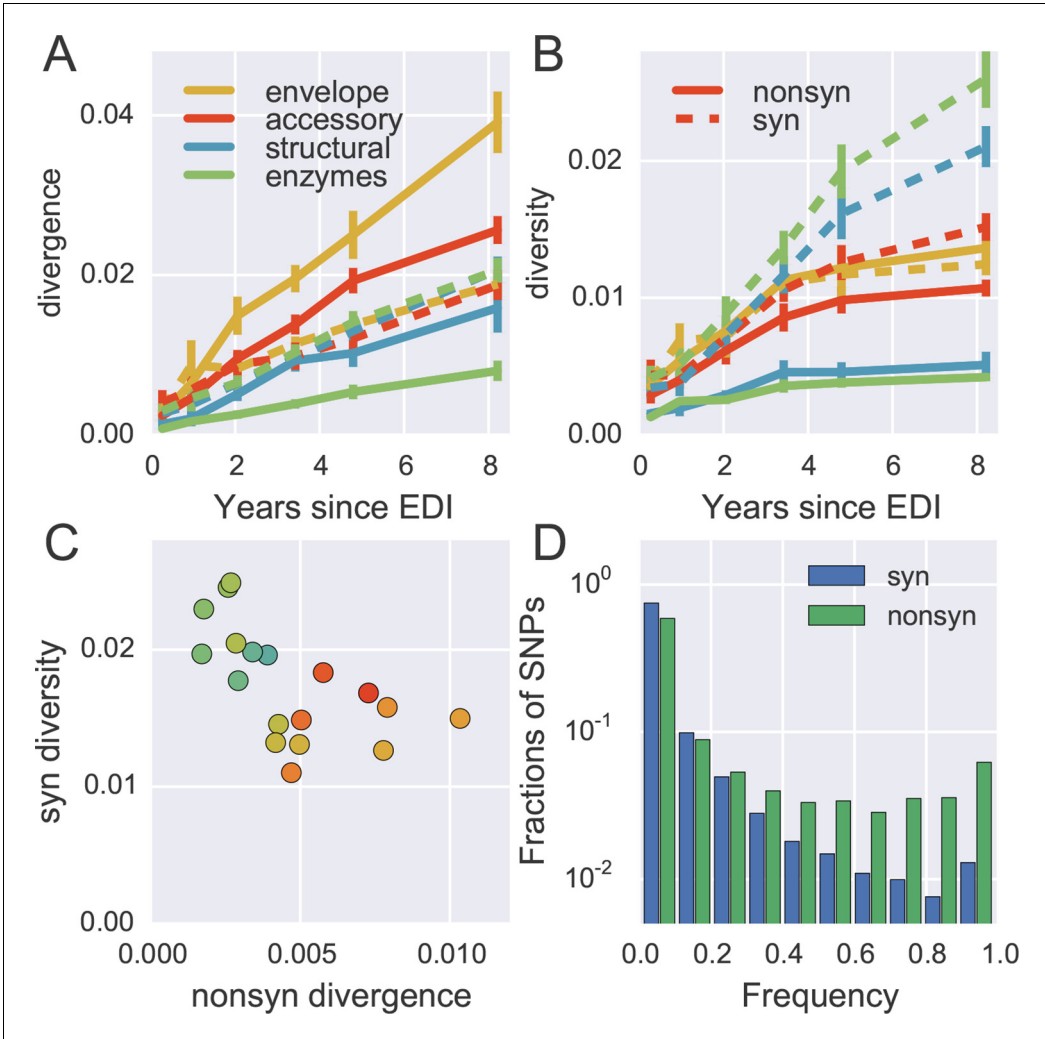

**Figure 5.** Distinct patterns of evolution across mutation types and regions. (**A**) shows divergence at nonsynonymous (solid) and synonymous (dashed) positions over time for different genomic regions averaged over all patients, measured as average Hamming distance from founder. While synonymous divergence is very similar in different regions, nonsynonyous divergence varies. (**B**) shows diversity through time, measured as average pairwise Hamming distance. Regions with high nonsynonymous diversity (and divergence) tend to have low synonymous diversity. Error bars represent standard deviations of patient bootstrap replicates. (**C**) shows the anti-correlation between the rate of nonsynonymous divergence and synonymous diversity in 1kb windows across the genome (color indicates position on the genome blue→green→yellow→red). (**D**) shows the site frequency spectrum of synonymous (blue) and nonsynonymous (green) SNPs. EDI, estimated date of infection; SNP, single nucleotide polymorphism.

The following source data is available for figure 5:

**Source data 1.** Tab-delimited files with plotted data.

the hosts (*Kawashima et al., 2009*; *Palmer et al., 2013*). In a diverse population of hosts, the most common state at a specific site is likely the preferred state, while rare alleles tend to be escape mutations that reduce viral replicative capacity (*Carlson et al., 2014*).

To quantify patterns of reversion and fitness cost, we classified sites in the approximate founder sequence of the viral populations in each subject as being identical or different from the HIV-1 group M consensus. *Figure 6A* shows the fraction of sites where the founder nucleotide is replaced by a mutant during the infection. This fraction is about ten-fold higher if the founder nucleotide differs

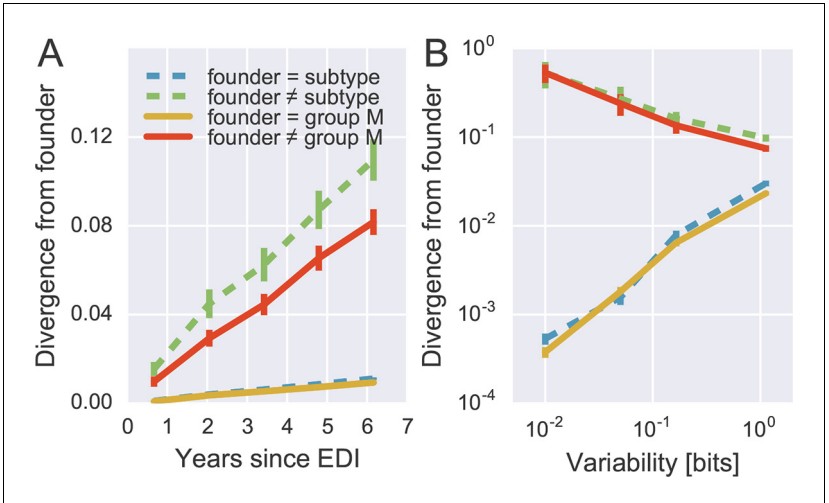

**Figure 6.** Rapid reversion at conserved sites. (**A**) Sites where the founder sequence differed from the subtype or group M consensus (upper curves) diverged about tenfold more rapidly than sites that initially agreed with the consensus (lower curves). (**B**) The rate of reversion increased with conservation (lower variability), while divergence away from consensus showed the opposite behavior (divergence is measured at 5−6 years). Error bars report the standard deviation of patient bootstraps. *Figure 6—figure supplement 1* shows the corresponding figure for amino acids rather than nucleotides.

The following source data and figure supplements are available for figure 6:

**Source data 1.** Tab delimited files with plotted data.

**Figure supplement 1—Source data 1.** Tab-delimited files with plotted data.

**Figure supplement 1.** Patterns of reversion at the amino acid level are similar to those at the nucleotide level.

from the group M consensus than if it is identical to the group M consensus. Reversion towards group M consensus occurs at a roughly constant rate throughout the observation time (5–8 years).

Of all changes accumulated by the viral populations, $30 \pm 2.5\%$ are reversions towards group M consensus (mean and standard deviation of patient bootstraps after 4–7 years). Similar results are found when comparing with the subtype consensus of each patient virus ($24 \pm 2.5\%$). Reversions are between 4 and 5 times more frequent than expected in the absence of a reversion bias (7.8 and 4.5%, respectively). These findings agree with results by *Allen et al. (2005),* who report that about 20% of amino acid substitutions are reversions.

By focusing on sites where the founder virus differed from the group M consensus, we are predominantly looking at weakly conserved sites. To control for conservation, we carried out the same analysis after stratifying sites by overall level of conservation. *Figure 6B* shows the result of this analysis, focusing on samples after 5–6 years for the sake of clarity. We find that the rate of reversion is highest at the most conserved sites. Almost 50% of all non-consensus positions at highly conserved sites had reverted to consensus after about 5 years – an almost 1000-fold excess. Even at the least conserved sites, divergence towards group M consensus exceeded divergence away from group M consensus by a factor of 3. These results suggest that the global HIV-1 group M consensus sequence represents an 'optimal' HIV-1 sequence, which acts as an attractor for the evolutionary dynamics within hosts. This attraction is strongest at conserved sites, but extends to the least conserved sites in the genome.

## Lack of long-range linkage due to frequent recombination

To quantify the decoupling of SNPs by recombination, we calculated linkage disequilibrium (LD) between SNPs as a function of distance for each of the six fragments, see *Figure 7*. For most fragments, we observed a consistent decrease of LD over the first 100–200 bps, with fragment 5 being

an exception with linkage of mutations at longer distances. Importantly, our linkage control (a 50/50 mix of two distinct virus isolates and a total of 1250 RNA molecules per PCR fragment) shows no decay of LD with distance, suggesting negligible RT-PCR recombination.

The observed decay of LD in patient samples is consistent with a recombination rate of $10^{-5}/bp/day$ as estimated in (*Batorsky et al., 2011*; *Neher and Leitner, 2010*). Our reasoning proceeds as follows. *Figure 5B* indicates that diversity accumulates over a time frame of 2–4 years, i.e., about 1000 days. Recombination at a rate of $10^{-5}/bp/day$ hits a genome on average every 100 bps in 1000 days. Mutations further apart than 100 bps are hence often separated by recombination and retain little linkage consistent with the observed decay length in *Figure 7*. The longer linkage in fragment 5 (env) might have several reasons that extend beyond our simple argument: (i) homologous recombination might be suppressed in the most variable regions, (ii) the accuracy of SNP frequency estimates is lower in F5 due to poorer amplification, and (iii) the rapid evolution of env due frequent substitutions and sweeps gives less time to break up linkage. In particular, as shown in *Figure 5C*, frequent and strong selective sweeps affect synonymous diversity in physical proximity along the genome, confirming the presence of linkage at short distances.

For phylogenetic analysis, we can extract haplotypes from the sequencing reads up to 500 bp in length. Only in the more diverse regions are 500 bp sufficient for well-resolved phylogenies (see *Figure 8*). However, we find that linkage does not extend beyond 100–200 bp. Hence the read length is not a limiting factor. Only during rapid population shifts such as drug resistance evolution, long read technologies such as PacBio would be necessary to capture the evolutionary dynamics (*Nijhuis et al., 1998*).

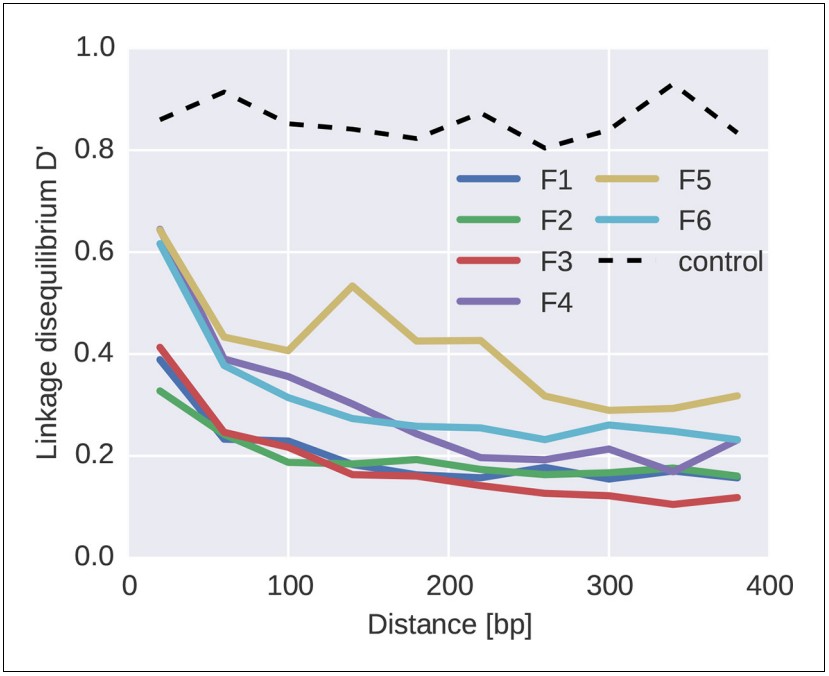

**Figure 7.** Linkage and recombination. Linkage disequilibrium decays rapidly with distance between SNPs. Colored lines correspond to the different fragments, each averaged over patients. The dashed line shows data from a control experiment for PCR recombination, where two cultured virus populations were mixed. No PCR recombination is observed.

The following source data is available for figure 7:

**Source data 1.** Tab-delimited files with plotted data.

# Discussion

We have presented a comprehensive portrait of intrapatient evolution of HIV-1 that covers almost the entire genome of the virus, characterizes minor genetic variants, and tracks the fate and dynamics of these variants over a follow-up period of up to 8 years in nine patients. We find that, during the infection, HIV-1 explores the sequence space surrounding the founder virus systematically; similar mutational patterns are observed within different, unrelated patients. Linkage between mutations is limited to approximately 100 bp, so the virus population can accumulate substitutions independently in different regions of the genome as suggested by theoretical models (*Mostowy et al., 2011*; *Rouzine and Coffin, 2005*). Nonetheless, local dynamics of SNPs is often dominated by hitchhiking between neighboring mutations, resulting in an anticorrelation between nonsynonymous divergence and synonymous diversity. A large fraction of all substitutions are reversions towards the global HIV-1 consensus sequence, and these reversions steadily accumulate throughout infection.

The evolutionary dynamics of HIV-1 populations is the result of stochastic forces like mutation and frequent bottlenecks, deterministic fixation of favorable mutations, and recombination. The relative importance of these forces remains unclear (*Brown, 1997*; *Frost et al., 2000*; *Kouyos et al., 2006*; *Maldarelli et al., 2013*; *Pennings et al., 2014*; *Rouzine and Coffin, 1999*). Our observation that intrapatient diversity recapitulates diversity seen across HIV-1 group M and the strong tendency to revert towards consensus suggest that, in chronic infection, selection determines diversity. The reproducible exploration of sequence space can coexist with frequent adaptation only in frequently recombining large populations (*Neher et al., 2013*). We observe that mutations further apart than 100 bp are effectively shuffled by recombination in most parts of genome, consistent with previous estimates of the HIV-1 recombination rate (*Batorsky et al., 2011*; *Neher and Leitner, 2010*). Linkage and stochastic effects become stronger with increasing frequency of strength of selection, consistent with lower synonymous diversity and more LD in env.

While rapid CTL escape at 5–10 sites over the first 2 years of infection has been documented in detail (*Allen et al., 2005*; *Goonetilleke et al., 2009*; *Herbeck et al., 2006*; *Jones et al., 2004*; *Liu et al., 2011*) and population level associations between specific HLA types and escape variants suggest widespread CTL escape (*Kawashima et al., 2009*), the effect of escape and reversion on long-term evolutionary trends is less clear (*Lythgoe and Fraser, 2012*; *Roberts et al., 2015*). We find a strong tendency for viral populations to revert towards the global HIV-1 consensus. At sites where the founder sequence differs from the subtype consensus, substitutions are almost five-fold overrepresented: Instead of ≈5% reversions expected based on the fraction of sequence at which the founder virus differs from consensus of the HIV-1 subtype, almost 25% of substitutions are reversions, in agreement with earlier reports on reversion of CTL escapes (*Allen et al., 2005*; *Li et al., 2007*). This tendency to revert increases with the level of conservation of the site, suggesting a quantitative relationship between fitness cost and conservation. While reversion is particularly prevalent in acute infection (*Li et al., 2007*), we show that reversion is not limited to early infection but happens throughout chronic infection.

The bias towards reversion results in a two- to three-fold reduction of the long-term evolutionary rate of HIV, a trend that is reinforced by selection during transmission (*Carlson et al., 2014*; *Sagar et al., 2009*). Inter-individual evolutionary rates of HIV-1 are two to six times lower than intra-individual rates, and a number of possible mechanisms have been suggested to explain this discrepancy (*Lythgoe and Fraser, 2012*). Our results strongly indicate that most of this mismatch can be explained by steady reversion during infection; other factors such as retrieval of 'stored' latent variants or stage-specific selection might also contribute to the rate mismatch (*Immonen and Leitner, 2014*; *Lythgoe and Fraser, 2012*).

The high rate of reversion has implications for phylogenetic dating. Given the five-fold excess of reverting minor variation, reversion would balance divergence once the typical distance from the consensus sequence equals 17%, corresponding to a nucleotide diversity of about 30%; this is remarkably close to the actual divergence between HIV-1 groups M, N and O (*Li et al., 2015*). On longer distances, this simple argument will have to be modified due to compensatory mutations resulting in gradual shift of the preferred state at some positions; nonetheless, it indicates a dramatic slowing down of divergence at a scale of the HIV-1-SIVcpz divergence. This apparent deceleration of evolution could explain the contradictory findings of attempts to date the age of HIV-1 and SIV (*Worobey et al., 2010*). The strong and lasting preference for specific nucleotides needs to be

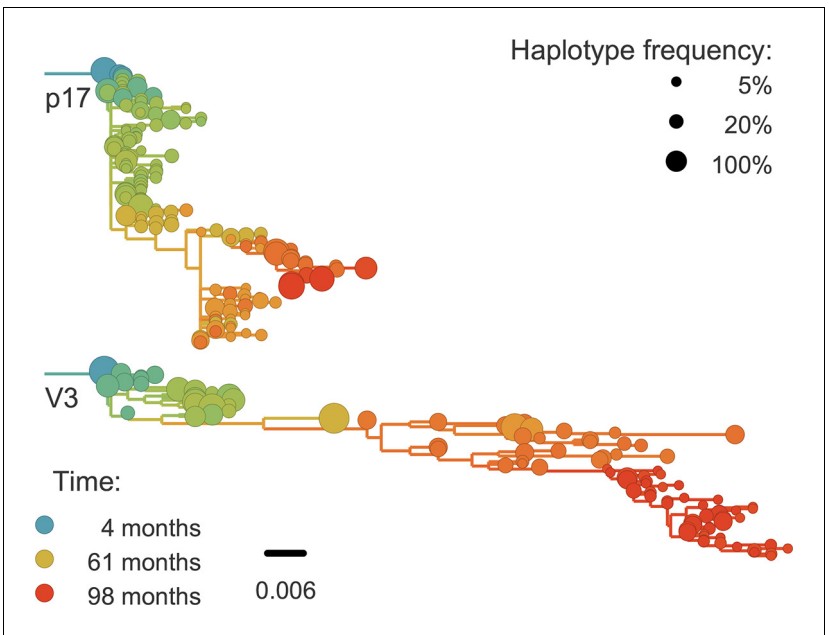

**Figure 8.** Phylogenetic trees of minor genetic variants. In rapidly evolving genomic regions, trees that include minor genetic variants (haplotypes) approximate the true phylogeny. Here p17 in gag and the variable loop 3 in env from patient p1 are compared; many more trees are available on the website. Trees are reconstructed using FastTree (*Price et al., 2009*).

accounted for in phylogenetic analysis, as has recently been shown using experimentally determined fitness landscapes of influenza virus proteins (*Bloom, 2014*).

Our observations are consistent with results by *Ashenberg et al. (2013)*; *Doud et al. (2015),* who showed that amino acid preferences are mostly conserved between related influenza strains. Similarly, divergence between HIV-1 subtypes is small enough that epistatic interactions have not yet changed the majority of the preferred states. With increasing evolutionary distance between strains, the molecular context and preferred amino acid at a particular position is more and more likely to differ (*Pollock et al., 2012*). Nevertheless, a subset of amino acids preferences are conserved almost universally (*Risso et al., 2015*) and enable sequence based homology search.

The concordance between intrahost variation and patterns of conservation across HIV-1 group M hints at universal fitness costs of mutations. Recently, cross-sectional conservation has been used as a proxy for fitness costs in models of HIV-1 fitness landscapes (*Dahirel et al., 2011*; *Ferguson et al., 2013*; *Mann et al., 2014*). Since reproducible intrapatient diversity likely reflects fitness costs of mutations in vivo, our results provide a direct justification for this approach. However, nine patients are insufficient to extend this analysis to fitness interactions between mutations.

One limitation of this study was the availability of samples from patients with sufficiently long follow-up without therapy after a well-defined time of infection. The majority of patients were MSM infected with subtype B virus. Thus, we cannot exclude that the aspects of HIV-1 evolution that we have investigated may differ between transmission routes or HIV-1 subtypes. While substitution and recombination errors of our optimized protocol for HIV-1 RNA extraction and an RT-PCR are low, the other main limitation was that the number of available template molecules was small in some samples (see also *Iyer et al., 2015*). In principle, the Primer ID method, which labels and resequences each individual template, allows quantification of templates and almost complete elimination of experimental substitution and recombination errors (*Jabara et al., 2011*). However, we are not aware of a Primer ID protocol for genome-wide sequencing, which was essential to our study.

The analysis and means of data sharing of next-generation sequencing data from viral populations is still in its infancy. Raw reads require substantial post-processing before the data can be used to answer biological questions and technology is changing rapidly such that no standardized pipelines exist. To facilitate the exploration and further analysis of our data, we have developed a web server

that allows to browse characteristics of individual patients, e.g., graph the frequencies of single nucleotide polymorphism through time, and download different aspects of the data in convenient format. It is, for example, possible to select an arbitrary region of the genome with length <500 bp, such as the frequently investigated V3 region, and extract an alignment of haplotypes covering this region along with their frequency from all time points belonging to one patient. We hope that the convenient access to processed data will facilitate follow-up studies on other aspects of viral evolution.

HIV-1 and other microbial populations evolve in a constant struggle between adaptation to a changing environment and maintenance of functionality. Large mutation rates and population sizes generate standing genetic diversity that is limited by the fitness costs rather than mutation rates. Hence the limiting factor for adaptation is not generating the useful mutations, but combining multiple mutations necessary to survive – e.g. escape mutations and reversions – and pruning deleterious mutations. In HIV-1, this process is facilitated by frequent recombination. We expect that the systematic exploration of sequence space, the reproducible patterns of minor variation, and frequent reversion will be characteristic of other RNA viruses. Properties of linkage between mutation will differ since mechanisms of recombination are diverse. But even though selective forces, recombination, and time scales will vary among different microbial populations, theoretical models of rapid adaptation population have shown that many features of the evolutionary dynamics are independent of the specific system (*Fisher, 2013*; *Neher, 2013*). Intrapatient evolution of HIV-1 is a unique opportunity to study this evolutionary dynamics directly in vivo.

## Materials and methods

### Patient selection and samples

The inclusion criteria for the study patients were: a) a relatively well-defined time of infection based on a negative HIV antibody test less than 2 years before a first positive test or a laboratory documented primary HIV infection; b) no antiretroviral therapy (ART) for a minimum of approximately 5 years following diagnosis; and c) availability of biobank plasma samples covering this time period. The patients were selected from the Venhälsan HIV clinic in Stockholm, Sweden and were diagnosed between 1990 and 2003. Seven to twelve plasma samples per patient (200–1000 $\mu$l) were retrieved from the biobanks of the Karolinska University Hospital and the Public Health Agency of Sweden. These samples had been stored at -70°C following routine HIV RNA quantification. Information about the patients and the samples are summarized in *Supplementary file 1*. Results from routine HIV antibody tests, HIV antigen tests, plasma HIV RNA levels and CD4 counts were collected from the patient records. Four-digit HLA typing of HLA class I loci A, B, C and class II loci DR and DQ was performed at the laboratory of immunpathology in Tübingen University Hospital.

In the control experiments, we used HIV DNA from the following plasmids; NL4-3 (subtype B, DNA concentration 110 ng/$\mu$l corresponding to $1.35 \times 10^{10}$ copies/$\mu$l), SF162 (subtype B, DNA concentration 117 ng/$\mu$l corresponding to $1.43 \times 10^{10}$ copies/$\mu$l), pZM246F_10 (subtype C, DNA concentration 101 ng/$\mu$l corresponding to $1.35 \times 10^9$ copies/$\mu$l).

We also used HIV RNA from the following virus isolates LAI III (subtype B, 7500 copies/$\mu$l), 38540 (subtype C, 225,000 copies/$\mu$l) and 38304 (subtype B, 45,000 copies/$\mu$l).

### Time since EDI

The patients were classified according to the Fiebig staging system for primary and early HIV infection (*Fiebig et al., 2003*). In addition, we performed BED tests (Aware BED EIA HIV-1 Incidence Test, Calypte Biomedical Corporation, Portland, OR). For each patient, the date of infection was estimated using results from laboratory tests according to the following hierarchical scheme.

1. Laboratory confirmed symptomatic primary HIV infection (PHI). Infection was considered to have occurred 17 days prior to date of first hospital visit based on an average incubation time from infection to development of PHI of 14 days (*Gaines et al., 1988*) and an estimated patient delay of 3 days. This was applicable to patients no. 6 and 10.
2. Fiebig staging (*Fiebig et al., 2003*) was used if the necessary laboratory results were available (primarily HIV screening test and Western blot) and the patient found to be in Fiebig stage I–

V. Fiebig stages I–VI were considered to correspond to 13, 18, 22, 27, 65 and >100 days since infection based on *Cohen et al. (2011)*. This was applicable to patients no. 2 and 8.

3. For patients in Fiebig stage VI, we calculated time since EDI using a published time-continuous model of development of antibodies reactive in the BED assay (*Skar et al., 2013*). However, time since EDI was not considered to be <100 days for patients in Fiebig stage VI. This was applicable to patients no. 1, 3, 5, 9 and 11.

Comprehensive information about the data used for EDI determinations is provided in *Supplementary file 3*.

## Primer design

Primers were designed to cover almost the full HIV genome in six overlapping fragments, called fragments F1–F6 as illustrated in *Figure 1*. This allowed sequencing of nucleotide positions 571–9567 in the HxB2 reference sequence according to the Sequence Locator Tool available at www.hiv.lanl.gov. Because of the redundancy of the long terminal repeats (LTRs), this means that all genomic regions except positions 482-571 in the R region of the LTRs were sequenced. Primer design was performed using the subtype reference alignment and the PrimerDesign software available at www.hiv.lanl.gov (*Brodin et al., 2013*). PrimerDesign was used to find candidate forward and reverse primers targeting highly conserved regions of the HIV genome, with similar melting temperatures, and with minimal tendency for hairpin and primer-dimer formation. Candidate primers were manually adjusted if needed, tested and sometimes redesigned. For each genome fragment, both outer PCR primers and nested, inner primers were designed; inner primers were only used for template quantification and internal testing purposes. Alternative primer sets were created for genome fragments F3 and F5 because the PCRs with the original primers sometimes were inefficient. For fragment F5, the amplification problem was not completely alleviated despite trials with several different primer pairs. We believe that this might be be due to the extensive secondary structure in the RRE region. The primers are presented in *Supplementary file 2*. All primer positions except the 5' primer of F1 and the 3' primer of F6 are contained in neighboring amplicons and hence sequenced. The primer part of the reads itself was trimmed after sequencing (see below).

## RNA extraction and amplification

For each sample, 400 $\mu$l of plasma (if available) was divided into two 200 $\mu$l aliquots. Total RNA was extracted using RNeasy Lipid Tissue Mini Kit (Qiagen Cat. No. 74804). Each aliquot was eluted twice with 50 $\mu$l RNase free water to maximize HIV RNA recovery. The four eluates were pooled, giving a total volume of 200 $\mu$l of RNA per sample. The RNA was divided into twelve 14 $\mu$l aliquots for duplicate one-step RT-PCR with the outer primers for fragments F1 to F6 and Superscript III One-Step RT-PCR with Platinum Taq High Fidelity Enzyme Mix (Invitrogen, Carlsbad, California, US).

Remaining RNA was used for template quantification (see below). The one-step RT-PCR was started with cDNA synthesis at 50°C for 30 min and denaturation step at 94°C for 2 min followed by 30 PCR cycles of denaturation at 94°C for 15 s, annealing at 50°C for 30 s and extension at 68°C for 90 s and a final extension step at 68°C for 5 min. A second nested PCR was used for template quantification and in some of the control experiments. For the second PCR, 2.5 $\mu$l of the product from the first PCR was amplified with Platinum Taq High Fidelity. The second PCR consisted of a denaturation step at 94°C for 2 min, followed by 30 PCR cycles with denaturation at 94°C for 15 s, annealing at 50°C for 20 s and extension at 72°C for 90 s and a final extension at 72°C for 6 min. Other PCR conditions were also tried during assay development.

After PCR, the duplicate amplicons from each of the six overlapping PCRs were pooled and purified with Illustra GFX PCR DNA and Gel Band Purification Kit (28-9034-70, VWR) or AGENCOURT AMPure XP PCR purification kit (A63881, Beckman Coulter AB). Purified amplicons from each sample were quantified with Qubit assays (Q32851, Life Technologies) and thereafter diluted and pooled in equimolar concentrations.

## DNA library preparation

The Illumina Nextera XT library preparation protocol and kit were used to produce DNA libraries. The original protocol was optimized for longer reads and amplicon input in the following fashion: (i) input DNA concentration to tagmentation was increased to 0.3 ng/$\mu$l to reduce overtagmentation;

(ii) the number of post-tagmentation PCR cycles was raised up to 14 for samples with very low input DNA; (iii) post-PCR purification was done using Qiagen Qiaquick columns to maximize large-inserts throughput as compared to magnetic-bead based protocols; (iv) sze selection was performed using the SageScience BluePippin system with 1.5% agarose gel cassettes and internal marker R2, selecting sizes of 550–900 bp (including dual index Nextera adapters, final insert sizes 400–700 bp); (v) size-selected eluates were pooled, buffer-exchanged into EB (10 mM Tris-HCl), and reconcentrated to at least 2 nM.

## Sequencing

The Illumina MiSeq instrument with 2 × 2 bp or 2 × 2 bp sequencing kits (MS-102-2003/MS-102-3003) was used to sequence the DNA libraries. We performed 26 paired-end sequencing runs. Overall, we obtained around 200 Gbases of output, i.e., 300 Mbases for each PCR amplicon. The median number of reads per amplicon was 80,000 (quartiles 20,000–220,000, max 2 millions). All read files have been uploaded to ENA with study accession number PRJEB9618.

## Read mapping and filtering

Bases were called from the raw images using Casava 1.8. The reads were analyzed from that point on using a custom pipeline written in Python 2.7 and C++. We favored this pipeline over existing programs because HIV-1 is a diverse species and both coverage and genetic diversity typically fluctuate by many orders of magnitude across the genome. The pipeline works as follows: (i) reads were mapped onto the HIV-1 reference HxB2, using the probabilistic mapper Stampy (*Lunter and Goodson, 2011*); (ii) mapped reads were classified into one of the six overlapping fragments used for RT-PCR (ambiguous and chimeric reads were discarded), and trimmed for PHRED quality above or equal 30 except for one isolated position per read; RT-PCR primers were also trimmed at this step; (iii) a consensus sequence was computed for each fragment in each sample from a subset of the reads, using a chain of overlapping local multiple sequence alignments (each covering around 150 bp); (iv) the reads were re-mapped, this time against their own consensus; (vii) genetic distance from the consensus were computed, and reads with a distance higher than a sample- and fragment-specific threshold were discarded. Each threshold, calculated to exclude even traces of cross-contamination that might have happened during RNA extraction, PCR amplification, or library preparation, was established by plotting the distribution of Hamming distances of reads from the sample consensus, and excluding reads that are further away than the tail of the main peak. Contaminations appeared as a second peak at higher distances, recombinants as a fat tail: both were excluded. Reads were also trimmed for mapping errors at the edges (small indels); (viii) filtered reads were mapped a third time against a patient-specific reference that was as similar as possible to the consensus sequence from the earliest time point; (ix) reads were re-filtered and checked again for cross-contamination. The tree of minor variants of all patients in *Figure 1—figure supplement 1* shows the absence of cross-contamination. The pipeline was equipped with extensive consistency checks for base quality, mapping errors, and contamination, and is based on the open source projects numpy (*van der Walt et al., 2011*), matplotlib (*Hunter, 2007*), Biopython (*Cock et al., 2009*), samtools (*Li et al., 2009*), pandas (*McKinney, 2011*), and SeqAn (*Döring et al., 2008*).

## Quality controls

### PCR and sequencing errors

We added 1–5% of PhiX control DNA to all our sequencing runs. Comparison of reads that mapped to PhiX with the PhiX reference sequence revealed that the Phred quality score were a rather reliable indicator of error rates, with saturation around 0.1%. SNP calls were calculated from the reads using a custom pile-up restricted to reads with a PHRED quality >=30 corresponding to a minimal theoretical error rate of 0.1% per base. However, in a dedicated set of control experiments, we established that the main source of error is PCR, see *Figure 1* in the main text.

We estimated the error rate of the PCR and sequencing pipeline by amplifying and sequencing plasmid clones (NL4-3, SF162, LAI-III). In most of these experiments, we used $10^4$ template molecules. The distribution of SNP frequencies found in the sequencing reads generated from the plasmid were compared with sequencing results from a typical patient sample, see *Figure 1*. After quality filtering, PCR and sequencing errors never exceed 0.3% of reads covering a particular

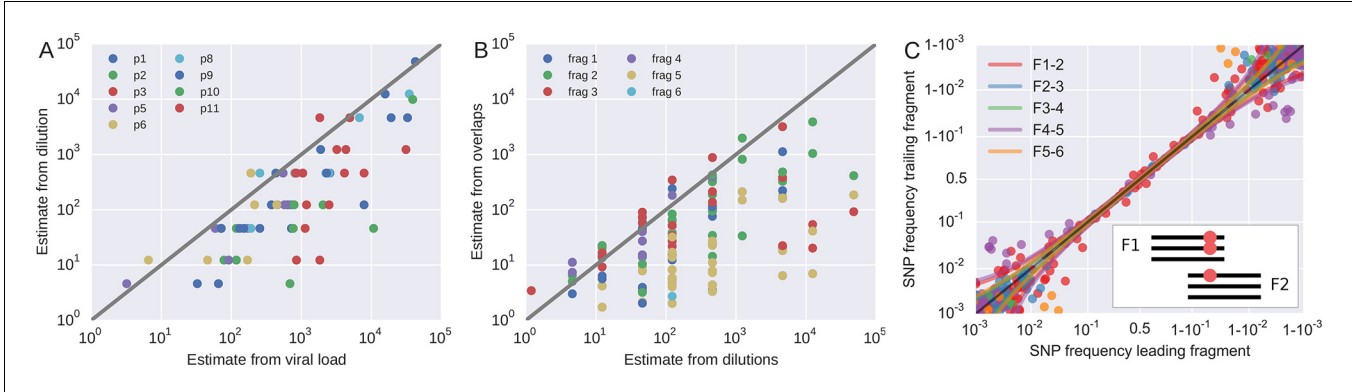

**Figure 9.** Template quantification and accuracy of SNP frequency estimates. The left panel shows actual template numbers as experimentally determined by end point dilution (*y*-axis) vs. the maximum theoretical template input as estimated based on plasma virus levels. The estimates from the dilutions are typically a factor of three below the expectation from viral load, suggesting generally good template recovery. The center panel show the correlation of fragment-specific effective template numbers estimated from the concordance of SNPs in fragment overlaps. Again, the correlation is generally good, but fragment F5 often has estimates lower than those from the dilution series consistent with problems faced in amplification of this fragment. Information from fragment F6 is largely absent, since there is very little variation in the F5–F6 overlap. The right panel plots SNP frequencies in overlaps measured in the leading fragment against the trailing fragment. Deviations from the diagonal are due PCR bias and random sampling from limited template molecules. Given enough diversity in overlap regions, the concordance of variant frequencies can be used to estimate fragment specific template input and accuracy of SNP frequencies. SNP frequencies in overlaps of the individual fragment pairs are shown in *Figure 9—figure supplement 2*.

The following figure supplements are available for figure 9:

**Figure supplement 1.** Concordance between two samples 30 days apart (samples 4 and 5 of p1).

**Figure supplement 2.** This figure shows the same data as *figure 9C*, but different overlaps are shown in separate panels.

position. Hence SNPs found in more than 0.3% of all reads likely represent biological variation. It is noteworthy that this neither implies that we detect all biological variants at frequencies above 0.3% nor does the frequency of a SNP among the reads necessarily reflect the frequency of that SNP in the sample. For many samples, the detection limit and accuracy of quantification of minor variants was not limited by the sequencing error, but by the RNA template input and potential PCR bias.

## Template input

To roughly quantify the actual number of HIV RNA templates that were subjected to sequencing, we analyzed a dilution series of the RNA (1:10, 1:100, 1:1,000, and 1:10,000) with the nested primers for fragment F4 using the PCR conditions described above. Tests were done in duplicate on the same plate as amplifications for sequencing and the amplicons were visualized by agarose gel electrophoresis. Templates numbers were estimated based on the assumption of Poisson sampling. Thus, for each sample, this dilution series provides an estimate of the number of templates that make it into the sequencing library. The results from the limiting dilution experiments correlated well (rank correlation p = 0.7) with routine plasma HIV-1 RNA level measurements that had been performed at time of sampling, see *Figure 9A*. These comparisons showed that the median template cDN recovery efficiency was 30% (quartiles 17–60%), which should be regarded as satisfactory since the amplicons were relatively long and the samples had been stored for up to 24 years and sometimes also freeze-thawed prior to this study.

## PCR bias

In addition to quantification of template input for each sample, we separately estimated the recovery of sequence variation in each of the six sequenced amplicons. Adjacent fragments overlap each other by a few hundred bases. Variation in the overlapping regions is amplified and sequenced on both the leading and trailing fragment completely independently and the concordance of the

observed variant frequencies, see *Figure 9*, can be used to detect PCR amplification bias or poor recovery in a subset of the fragments.

If deviations of SNP frequencies are due to random sampling from the pool of RNA molecules and stochastic PCR efficiency, the variance between the frequencies observed on two neighboring fragments should be $x(1-x)(n_L^{-1} + n_T^{-1})$, where $x$ is the true frequency of the SNP. The numbers $n_L$ and $n_T$ correspond to effective template numbers specific to leading (*L*) and trailing (*T*) fragment. For each SNP in the overlap, we calculate the difference $\delta$ and the mean $x$ of the frequencies reported on the *L/T* fragment. An estimate of $n_L^{-1} + n_T^{-1}$ is then given by

$$\frac{1}{k+p}\left[\sum_{i=1}^{k}\frac{\delta_i^2}{x_i(1-x_i)} + \frac{p}{n_{dil}}\right] \quad , \tag{1}$$

where $k$ is the number of SNPs in the overlap. To increase robustness of this estimate, we include $p$ pseudo counts each of which contributes $n_{dil}^{-1}$, where $n_{dil}$ is the template number estimated from the dilution series. Since we have six fragments but (in the best case) only five overlaps, the effective fragment template numbers are not uniquely determined. However, their inverse has to be smaller than the minimum of the estimate of $n_L^{-1} + n_T^{-1}$ at any overlap the fragment participates in. Hence, we assign a lower bound for the fragment-specific template estimate by using the inverse of this minimum, provided we find at least $k = 5$ SNPs in the overlap at an average frequency above 0.03. We used $p = 3$. Changing the parameters does not change the numbers qualitatively.

The fragment specific estimates of template numbers are consistent with the estimates based on limiting dilution (see *Figure 9*, rank correlation around $p = 0.5$ for most of the fragments), but indicated that variant recovery in fragment F5 was sometimes poor, consistent with the difficulties we encountered when amplifying fragment F5. Furthermore, the agreement of SNP frequencies in the overlaps indicate that the primers were well-designed and did not induce significant primer-related PCR amplification biases. These fragment-specific estimates are only available for samples and fragments with sufficient diversity in the overlap regions and are often lacking for early samples and fragment F6. The latter is due to the very conserved overlap with fragment F5.

Taken together, these controls show that, depending on the sample and fragment, we can estimate frequencies of SNPs down to 1% accuracy (corresponding to several thousand effective templates). In some cases, however, the template number was low or recovery poor such that no more than presence absence of a variant could be called. All available fragment specific estimates are shown in *Figure 9B*. In addition to lack of concordance of SNP frequencies in overlaps, problematic fragments lack diversity or have a granular distribution of SNP frequencies such that they can be flagged and removed from specific analysis. The fact that PCR efficiency can vary within one sample is illustrated in *Figure 9—figure supplement 1* that compares SNP frequencies estimated in samples 4 and 5 of patient 1 that are only 30 days apart: While SNP frequencies agree to within expectation for five out of six fragments; fragment F5 shows strong deviations linked to suboptimal PCR efficiency. However, even in problematic samples, our amplification and sequencing approach does not generate spurious variation, as shown by the absence of diversity in early samples or conserved sites (see below).

## In vitro recombination and linkage

The paired-end reads obtained correspond to inserts of up to 700bp in length and therefore provide information on linkage of mutations up to that distance. However, cDNA synthesis and PCR have both the potential to generate in vitro recombination, and true biological linkage is preserved only if the frequency of in vitro recombination is low. To estimate the in vitro recombination in our experimental setup, virions from two subtype B HIV isolates, LAI III and 38304 (the latter obtained from a Swedish HIV-1 patient infected in Brazil), were mixed in equal concentrations. Aliquots of this mix (approximately 1250 RNA molecules per PCR fragment) were amplified with the six overlapping one-step RT-PCRs as described above in both single and nested PCR mode and then sequenced.

PCR recombination is known to occur predominantly at high amplicon concentrations due to heteroduplex formation of incompletely extended molecules (*Di Giallonardo et al., 2013*; *Mild et al., 2011*). Consistently, we observe no PCR recombination within the first PCR (which starting at low template input does not saturate), while we observe substantial PCR recombination during a second nested PCR. Since our library preparation protocol requires very low input DNA, we do not need

nested PCR, avoiding this source of PCR recombination. The fact that PCR recombination occurs during a second PCR shows that the two viruses used for this control (both subtype B) are similar enough that divergence does not interfere with heteroduplex formation. *Figure 7* includes the linkage disequilibrium observed in the control experiment as a function of distance, showing that linkage is high and not lost with distance during the first PCR.

## Analysis

Python scripts that generate each figure shown in the manuscript are available at github.com/neher-lab/HIVEVO_figures. In these scripts, all parameters, settings, and calculations are explicitly documented.

### Divergence rates across the HIV-1 genome

Average divergence defined as the fraction of non-founder alleles in a sliding window of 300bp was regressed against the estimated times since infection using a linear model without intercept. Different data points were weighted by the total divergence, reflecting the expected scaling of the variance if divergence is due to changes at many independent sites. The evolutionary rate was than mapped to the corresponding coordinate in the HxB2 reference.

### Phylogenetic analysis

To construct trees in different regions of the HIV-1 genome, we extracted reads covering the desired region, restricted them to haplotypes above the desired frequency threshold and constructed a maximum likelihood phylogeny using FastTree (*Price et al., 2009*). Trees were rooted at the consensus sequence of the earliest available sample.

### Linkage disequilibrium

To calculate linkage disequilibrium, we first constructed four-dimensional matrices for each fragment and each sample that report the number of times reads mapped such that a nucleotide $n_1$ at position $x_1$ was jointly observed with nucleotide $n_2$ at position $x_2$. These matrices, reduced to variable positions, are available on the website. From these matrices, we calculate the joint frequency $p_{12}$ ($x_1$, $x_2$) of the majority nucleotide at positions ($x_1$, $x_2$). The normalized linkage disequilibrium measure is then given by

$$D^{'} = \frac{p_{12} - p_1 p_2}{D_{max}} \qquad (2)$$

where $p_1, p_2$ are the frequencies of the majority nucleotide at the respective positions, and

$$D_{max} = \begin{cases} \min\Big(p_1 p_2, \ (1-p_1)(1-p_2)\Big) & p_{12} < p_1 p_2 \\ \min\Big(p_1(1-p_2), \ (1-p_1)p_2\Big) & p_{12} > p_1 p_2 \end{cases}$$

Values of $D^{'}$ are then binned by distance and plotted. To reduce noise, only SNPs at frequencies between 20 and 80% at positions with a coverage >200 were included. Furthermore, we required that the fragment-specific depth estimate exceeded 40 or the estimated template input exceeded 200. The latter is necessary to include fragment F6, since the overlap with fragment 5 often had insufficient diversity to allow for a fragment-specific depth estimate.

### Mutation classification and CTL epitope prediction

Synonymous and non-synonymous divergence was analyzed on a per site basis to avoid confounding by different numbers of transitions and transversions. Positions were classified as synonymous based on the following criterion: The consensus amino acid at this position did not change throughout infection and at least the transition at this site is synonymous. In this case, the observed mutations at this site are almost always synonymous.

The functional categories for genomic regions were the following: structural: p17, p24, p6, p7; enzymes: protease, RT, p15, integrase; accessory: vif, vpu, vpr, nef; envelope: gp120 and gp41. For the CTL epitope prediction, ranked epitope lists were obtained using the web service MHCi ( tools. immuneepitope.org/mhci). This tool uses several prediction methods ranked based on previously

observed performance: Consensus (*Moutaftsi et al., 2006*) > Artificial Neural Network (*Nielsen et al., 2003*) > Stabilized Matrix Method (*Peters and Sette, 2005*) > NetMHCpan (*Hoof et al., 2008*). For each patient, we submitted the consensus amino acid sequences of the viral proteins at the first time point together with 4-digit HLA types at MHC-I. To analyze putative CTL escape in our patients, we used the first 80 epitopes from the ranked list as a compromise between false negatives and false positives. This number maximized the statistical signature for HIV-1 substitutions being within epitopes, but similar results were obtained by taking the first 50-100 epitopes in the lists.

We preferred computational predictions over experimentally verified CTL epitopes, since the latter are incomplete and biased towards common virus sequences and HLA types. For each patient, we submitted the approximate founder sequence (consensus sequence of the first sample) to MHCi and obtained a list of peptides putatively presented by the HLA alleles (A/B/C) of the respective host ranked by prediction score.

We considered the parts of the HIV-1 genome that putatively is targeted in any of the nine patients, and asked whether the density of nonsynonymous substitutions is higher in the part that is predicted to be presented in a focal patient compared to the part that is not presented. We excluded the variable loops of gp120 and the external part of gp41 from this analysis to avoid confounding by antibody selection. Repeating the same analysis for synonymous mutations did not yield any enrichment as expected.

## Comparison with subtype diversity

Full genome HIV-1/SIVcpz sequences were downloaded from LANL HIV database. Sequences belonging to subtypes B, C, and 01_AE were individually aligned to HxB2 (using pairwise alignment functions from SeqAn (*Döring et al., 2008*)) to construct a summary of diversity within subtypes in HxB2 coordinates. Similarly, a group M diversity summary in HxB2 coordinates with made from sequences of subtypes A, B, C, D, F, G, H, restricted to at most 50 sequences from any given subtype. From these HxB2 indexed alignments, entropy was calculated columnwise and the consensus sequence determined as the majority nucleotide.

When comparing HIV variation within patients to cross-sectional diversity, only positions of the founder sequence were used that mapped to HxB2. Similarly, only positions in the cross-sectional alignment were used that had less than 5% gaps relative to HxB2. This effectively masked regions of the genome that frequently experience insertions of deletions.

To investigate divergence and reversion, intrapatient evolution was studied separately at sites where the approximate founder sequence agreed/disagreed with the respective subtype or group M consensus. Divergence was assessed on a per site basis.

### Website

The website was realized using the Flask web framework, Bootstrap from Twitter as a frontend engine and jQuery and D3.js as a JavaScript library for interactive plots (*Bostock, 2015*; *Ronacher, 2015*). The webpage is available at hiv.tuebingen.mpg.de.

## Acknowledgements

This work was supported by the European Research Council through grant Stg. 260686. We would like to thank Bianca Regenbogen and Diep Thi Ngoc Tran for help with the design of the web page and Thomas Leitner for helpful feedback and discussion. We would also like to express our gratitude to the study participants.

## Additional information

### Competing interests

RAN: Reviewing editor, *eLife.* The other authors declare that no competing interests exist.

## Funding

| Funder | Grant reference number | Author |
| --- | --- | --- |
| European Research Council | 260686 | Jan Albert<br>Richard A Neher |

The funders had no role in study design, data collection and interpretation, or the decision to submit the work for publication.

## Author contributions

FZ, JB, JA, Conception and design, Acquisition of data, Analysis and interpretation of data, Drafting or revising the article; LT, Acquisition of data, Analysis and interpretation of data; CL, Conception and design, Acquisition of data; GB, Conception and design, Contributed unpublished essential data or reagents; RAN, Conception and design, Analysis and interpretation of data, Drafting or revising the article

## Author ORCIDs

Richard A Neher, http://orcid.org/0000-0003-2525-1407

## Ethics

Human subjects: The study was carried out according to the Declaration of Helsinki. Ethical approval was granted by the Regional Ethical Review board in Stockholm, Sweden (Dnr 2012/505-31/12). Patients participating in the study gave written and oral informed consent to participate.

# Additional files

## Supplementary files

• Supplementary file 1. Data on viral load, CD4 count, date, and average coverage for all samples sequenced.

• Supplementary file 2. Primers used to amplify the HIV-1 genome in six overlapping fragments.

• Supplementary file 3. Data used to estimate the date of infection for all study participants.

## Major datasets

The following datasets were generated:

| Author(s) | Year | Dataset title | Dataset ID and/or URL | Database, license, and accessibility information |
| --- | --- | --- | --- | --- |
| Zanini F, Brodin J, Thebo L, Lanz C, Bratt G, Albert J, Neher RA | 2015 | Whole genome deep sequencing of HIV-1 | www.ebi.ac.uk/ena/data/view/PRJEB9618 | Publicly available at the EBI European Nucleotide Archive (Accession no: PRJEB9618). |
| Zanini F, Brodin J, Thebo L, Lanz C, Bratt G, Albert J, Neher RA | 2015 | Population genomics of intrapatient HIV-1 evolution | http://hiv.tuebingen.mpg.de | Interactive web application facilitating exploration and access to the data |

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
