## [Decision Letter]

Thank you for submitting your work entitled "Population genomics of intrapatient HIV-1 evolution" for consideration by *eLife*. Your article has been reviewed by three peer reviewers, and the evaluation has been overseen by a Reviewing Editor and Diethard Tautz as the Senior Editor.

The reviewers have discussed the reviews with one another and the Reviewing editor has drafted this decision to help you prepare a revised submission.

Summary:

In this paper, you use deep sequencing to track the genotypic evolution of HIV in individuals over multiple years. You observe patterns in the virus's evolution over time, and note a surprising amount of parallelism in the fate of specific mutations in the few patients you study. Although others have conducted similar studies, your study is more comprehensive than past efforts. Another major strength is that you develop a web-based visualization platform that enables both expert and non-expert users to inspect the results in detail. However, the current manuscript does not properly describe past efforts adequately, and this is the primary criticism that will need to be addressed before we can justify publication in *eLife*.

For example, in the Introduction, you may want to discuss the work described in the following references as part of the background and then make clear what is novel about your studies: Bernardin J Virol 2005; Bar KJ Plos Path 2012; Brockman J Virol 2010; Schneidewind A JVirol 2010. Also, Salazer-Gonzalez 2009, did not examine T cell escape in acute HIV infection; escape was examined Goonetilleke et al. 2009 (published in the same issue of JEM as Salazar-Gonalez) followed by Liu et al. JCI 2013. Key papers from the lab of P Borrow should also be recognized. Iyer S Plos One 2015 (lab of J Mullins) is a recent cross-sectional study in HIV-1 using deep sequencing techniques that is also relevant. Finally, the point made in the Discussion about how your results provide direct justification for past efforts to extract universal fitness landscapes of HIV from cross-sectional data might be considered for inclusion in the introduction as well (e.g., Dahirel, PNAS (2011); Fergusson, Immunity (2013); Mann PLOS CB (2014)).

In addition, it is important that you address the following comments in order to improve the paper:

Specific comments:

1) It is clear from Figure 1 that the major limitation for low-frequency SNPs is low template abundance, rather than sequencing errors. SNP frequencies are not expected to be accurate below 1/N where N is the number of template molecules, and N appears to be about 10 to 50. This would suggest that the SNP frequencies are not very accurate below about 2-10%. This idea seems to be consistent with the results in Figure 9, which show a very strong correlation between frequencies called from different fragments at moderate frequencies, but little correlation below about 1-5%. Although you do discuss this point to some degree, we think this point should be noted more emphatically. There is a great deal of interest in more accurate techniques for sequencing viruses, but as the results here make clear, there isn't really much benefit in improving sequencing accuracy if the accuracy is already > 1/N.

2) A major finding of the paper is the evidence for parallel evolution in the sense of similar positions accumulating similar divergence among patients, and of strong tendency of reversion to the same consensus identities in all strains. However, it is often unclear if this trend is occurring primarily on the protein level, primarily on the nucleotide level, or both. For instance, in Figure 4 and Figure 6 it is unclear if the trends are for nucleotide or amino-acid mutations – this should be clarified, and ideally results for both nucleotides and amino acids could be shown. Figure 3 might also benefit from showing similar data for both nucleotides and amino acids. In the Discussion you note, "The strong and lasting preference for specific nucleotides needs to be accounted for…". In fact, the cited reference discusses strong and lasting preferences for specific amino acids, not specific nucleotides.

3) How does the consistency of evolution between patients correlate with distance between those patients' founder viruses? A central claim in this study is that: "At a single nucleotide level, the spectrum of mutational possibilities is explored reproducibly." The evidence supporting this claim includes the correlation between the site-specific entropy of patient samples and natural sequences (Figure 4). However, something that is unclear from this comparison is whether variation is increasing at the same or different sites for different samples within this study. Can you make similar comparisons to Figure 4 across samples within the study? Are the correlations stronger between samples from individuals infected with genetically similar viruses?

4) More discussion is needed on the clinical characteristics of patients (viral load and CD4 count over time). Of note, participants had to be ARV naïve for at least 5 years, and thus some had a low viral load. How might factors leading to low viral load affect evolution? You note that some patients received ART during the study, and were included in an ongoing sub-study. However, it is not clear if the time points after ART initiation were included in the present study. If they were included, are there any interesting differences in evolutionary patterns between patients that received ART and patients that did not?

5) Five of nine subjects in the cohort were identified in either Fiebig V or VI. This staging precludes confident determination of the transmitted/founder viruses in these subjects and therefore the statement that “initial consensus sequence approximates the sequence of the founder virus(es)” is not supported. You may want to revise the terminology to reflect that, in these subjects, you deduced a consensus sequence of an early virus sequence.

Regarding the Fiebig staging and virus loads: These data are very confusingly described and presented. Was Fiebig staging performed during a visit prior to the available plasma samples tested? If so, when? Going through the supplementary file, the VL of the first sample available for sequencing is clearly outside the acute window for several of the patients described as detected in Fiebig II-IV. As above, how can you justify describing the first sequence as the founder? As shown by multiple groups, significant immune selection and recombination occurs within the first months of HIV infection.

6) You provide a convincing demonstration of reversal to global consensus and provide a discussion of the how this observation can be reconciled with the observed intra-patient and global HIV diversity. It might help the reader to have the discussion of the balance of divergence and reversal (presently in the fifth paragraph of the Discussion) sooner, e.g. in the subsection "Extensive reversion toward consensus". The findings in this regard relate to a major ongoing debate in molecular evolution about the extent to which diverged homologs retain strong propensities for specific amino acids. Pollock et al. (PNAS, 109:E1352) and Shah et al. (PNAS, 112:E3226) have argued that pervasive epistasis means that the propensities for specific amino acids means will diverge rapidly among homologs, which will quickly come to favor different amino acids at homologous sites. On the other hand, Ashenberg et al. (PNAS, 110:21071), Risso et al. (Mol Biol Evol, 32:440), and Doud et al. (Mol Biol Evol, doi:10.1093/molbev/msv167) have argued that the propensities for specific amino acids are similar among homologs. How do your results bear on this issue?

7) In the tree in Figure 1, near the yellow for patient 9 there is a black line that doesn't connect to any colored lines in the tree. What does this indicate?

8) It is confusing that there is one Methods section called "diversity and divergence" and another called "divergence and diversity".

9) The y-axis of Figure 5 should be labeled with the appropriate units, or this information should be added to the figure caption.

10) You state: "Figure 6 indicates that diversity accumulates over a time frame of 2-4 years." However, since Figure 6 shows the rate of reversion compared to variability at 5-6 years, it is not clear how this demonstrates that diversity accumulates over a 2-4 year period.

11) A*32 is overrepresented in this group; a much higher frequency than found in the broader Swedish population. Can the authors comment? Are these subjects in any way linked?

---

## [Author Response]

*In this paper, you use deep sequencing to track the genotypic evolution of HIV in individuals over multiple years. You observe patterns in the virus's evolution over time, and note a surprising amount of parallelism in the fate of specific mutations in the few patients you study. Although others have conducted similar studies, your study is more comprehensive than past efforts. Another major strength is that you develop a web-based visualization platform that enables both expert and non-expert users to inspect the results in detail. However, the current manuscript does not properly describe past efforts adequately, and this is the primary criticism that will need to be addressed before we can justify publication in eLife.*

Thank you very much for the constructive and helpful criticism, which has helped us to connect our findings better to existing literature and articulate more clearly the novel insights that can be obtained from longitudinal whole genome deep sequencing. While evolution during early infection or in specific regions of the HIV-1 genome have been extensively studied, we present for the first time a deep whole genome data set that allows quantification of the evolutionary dynamics throughout infection.

*For example, in the Introduction, you may want to discuss the work described in the following references as part of the background and then make clear what is novel about your studies: Bernardin J Virol 2005; Bar KJ Plos Path 2012; Brockman J Virol 2010; Schneidewind A JVirol 2010.*

We have expanded the part of the Introduction discussing prior studies of HIV evolution. Specifically, we have included the suggested references when discussion early CTL escapes, Nab escapes, fitness costs, and stability of compensated transmitted variants. All these references focus on patterns of evolution during the first few month after infection, while we study genome wide patterns of evolution from early to chronic infection.

*Also, Salazer-Gonzalez 2009, did not examine T cell escape in acute HIV infection; escape was examined Goonetilleke et al. 2009 (published in the same issue of JEM as Salazar-Gonalez) followed by Liu et al. JCI 2013.*

Our apologies, we mixed up these back-to-back papers. We now cite Goonetilleke et al. 2009 and Liu et al. 2013 when discussing early escape. We included the reference to Salazar-Gonzales et al. 2009 when discussing founder sequences.

*Key papers from the lab of P Borrow should also be recognized. Iyer S Plos One 2015 (lab of J Mullins) is a recent cross-sectional study in HIV-1 using deep sequencing techniques that is also relevant. Finally, the point made in the Discussion about how your results provide direct justification for past efforts to extract universal fitness landscapes of HIV from cross-sectional data might be considered for inclusion in the introduction as well (e.g., Dahirel, PNAS (2011); Fergusson, Immunity (2013); Mann PLOS CB (2014)).*

Thank you for these suggestions. We included a reference to work by the lab of P Borrow (Jones et al., 2004) when discussing previous work on CTL escapes in early infection. We further included a reference to Iyer et al. when discussing the necessity of quantifying templates and the limitations of next generation sequencing of diverse viral populations. We have added a brief discussion about the link between reversion, reproducible intrapatient diversity and inference of fitness landscapes from cross-sectional data. We also included the additional references as suggested.

*In addition, it is important that you address the following comments in order to improve the paper:*

*Specific comments: 1) It is clear from Figure 1 that the major limitation for low-frequency SNPs is low template abundance, rather than sequencing errors. SNP frequencies are not expected to be accurate below 1/N where N is the number of template molecules, and N appears to be about 10 to 50. This would suggest that the SNP frequencies are not very accurate below about 2-10%. This idea seems to be consistent with the results in Figure 9, which show a very strong correlation between frequencies called from different fragments at moderate frequencies, but little correlation below about 1-5%. Although you do discuss this point to some degree, we think this point should be noted more emphatically. There is a great deal of interest in more accurate techniques for sequencing viruses, but as the results here make clear, there isn't really much benefit in improving sequencing accuracy if the accuracy is already > 1/N.*

The number of input templates varies greatly from sample to sample. We now discuss the distribution of template input at greater length. The median template input per fragment is 120 with an interquartile range of 50-500. In most cases we have therefore more templates than indicated in Figure 1 and we can give meaningful SNP frequencies down to 1%. In some cases, template input limits accuracy more drastically.

The deviations in Figure 9 were largely due to fragment 5. We included an additional supplementary figure, Figure 9—figure supplement 2, that shows the correlation of SNP frequencies for individuals fragment overlaps. In overlaps that don't include fragment 5, SNP frequencies correlate well down to a few percent.

2) A major finding of the paper is the evidence for parallel evolution in the sense of similar positions accumulating similar divergence among patients, and of strong tendency of reversion to the same consensus identities in all strains. However, it is often unclear if this trend is occurring primarily on the protein level, primarily on the nucleotide level, or both. For instance, in Figure 4 and Figure 6 it is unclear if the trends are for nucleotide or amino-acid mutations – this should be clarified, and ideally results for both nucleotides and amino acids could be shown. Figure 3 might also benefit from showing similar data for both nucleotides and amino acids. In the Discussion you note, "The strong and lasting preference for specific nucleotides needs to be accounted for…". In fact, the cited reference discusses strong and lasting preferences for specific amino acids, not specific nucleotides.

We thank the reviewers for suggesting this analysis, which we also find to be an interesting point. We now provide Figure 3—figure supplement 2, Figure 4—figure supplement 1, and Figure 6—figure supplement 1 which show the results of the same analyses as in Figure 3, Figure 4, and 6 but at the amino acid level. Overall, we find that divergence along the genome (Figure 3, Figure 3—figure supplement 2), correlation with cross-sectional diversity (Figure 4, Figure 4—figure supplement 1), and patterns of reversions (Figure 6, Figure 6—figure supplement 1) all show similar dynamics at the amino acid and nucleotide levels. By comparing panel A in Figure 6—figure supplement 1 to Figure 6, it is evident that the overall scale of divergence is of course different for amino acids, but the relative dynamics at sites where the founder agrees or disagrees with cross-sectional consensus are the same as for nucleotides.

*3) How does the consistency of evolution between patients correlate with distance between those patients' founder viruses? A central claim in this study is that: "At a single nucleotide level, the spectrum of mutational possibilities is explored reproducibly." The evidence supporting this claim includes the correlation between the site-specific entropy of patient samples and natural sequences (Figure 4). However, something that is unclear from this comparison is whether variation is increasing at the same or different sites for different samples within this study. Can you make similar comparisons to Figure 4 across samples within the study? Are the correlations stronger between samples from individuals infected with genetically similar viruses?*

We added Figure 4—figure supplement 2 which shows that the correlation in nucleotide variation between between pairs of patients as a function of the genetic distance between the founder viruses of the patient pair. The genetic distance between founder viruses does not seem to have a big impact, but this conclusion is based essentially on two non-subtype B patients only, p1 and p6. More data is necessary for a quantitative study of the dependence of the similarity of intrapatient diversity on genetic distance.

*4) More discussion is needed on the clinical characteristics of patients (viral load and CD4 count over time). Of note, participants had to be ARV naïve for at least 5 years, and thus some had a low viral load. How might factors leading to low viral load affect evolution? You note that some patients received ART during the study, and were included in an ongoing sub-study. However, it is not clear if the time points after ART initiation were included in the present study. If they were included, are there any interesting differences in evolutionary patterns between patients that received ART and patients that did not?*

We have added more information about the samples, viral load, and CD4 count in the Results section. We have also included Figure 1—figure supplement 1, which shows viral load and CD4 counts over time. Previously these data were only shown on the website. It is true that the fact that patients were selected to have been treatment-naive for at least 5 years may have selected against rapid progressors. However, most patients were diagnosed in the 1990s, i.e. before the adoption of early start of ART. Thus, the last study CD4 count (before start of therapy) was relatively low for all patients; 340, 369, 140, 228, 287, 378, 158, 200, 251 for patients 1-3, 5-6 and 8-11, respectively. Information about this has been added to the Results section. We cannot exclude that rapid progressors might have somewhat different rate and pattern of HIV evolution.

All samples in this study were obtained before any start of ART. To avoid misunderstanding we have removed the mentioning of the substudy, which is not relevant to this manuscript (beginning of Results).

5) Five of nine subjects in the cohort were identified in either Fiebig V or VI. This staging precludes confident determination of the transmitted/founder viruses in these subjects and therefore the statement that “initial consensus sequence approximates the sequence of the founder virus(es)” is not supported. You may want to revise the terminology to reflect that, in these subjects, you deduced a consensus sequence of an early virus sequence.

Regarding the Fiebig staging and virus loads: These data are very confusingly described and presented. Was Fiebig staging performed during a visit prior to the available plasma samples tested? If so, when? Going through the supplementary file, the VL of the first sample available for sequencing is clearly outside the acute window for several of the patients described as detected in Fiebig II-IV. As above, how can you justify describing the first sequence as the founder? As shown by multiple groups, significant immune selection and recombination occurs within the first months of HIV infection.

It is true that our earliest sample is between one and seven month after infection and some mutations have likely spread through the population by that time. Hence we can only approximate, rather than exactly identify the founder sequence. However, the difference between the true founder and our approximation will be small compared to the evolution in the many years of follow-up after the first sample. All observation that we make in the manuscript should become even stronger if we knew the exact founder sequence. We added a discussion of the approximate nature of the inferred founder sequence to the manuscript in the subsection “Consistent evolution across the entire genome”.

We agree that the presentation of the Fiebig staging, virus loads and estimation of date of infection (EDI) was unclear. This includes the dates for sampling for the different tests (VL, Fiebig staging, BED and first sequence) which sometimes differed. For this reason we provide a new supplement table with comprehensive data, dates and comments, see [Supplementary-material SD9-data]. Please also note that the scheme for estimating date of infection has been revised to first consider information on date for a laboratory confirmed primary HIV infection. We have also run additional Western blots and BED tests. For these reasons the EDIs have changed for some patients. All analyses and graphs have been redone accordingly, but this has in no way changed the main results and conclusions.

*6) You provide a convincing demonstration of reversal to global consensus and provide a discussion of the how this observation can be reconciled with the observed intra-patient and global HIV diversity. It might help the reader to have the discussion of the balance of divergence and reversal (presently in the fifth paragraph of the Discussion) sooner, e.g. in the subsection "Extensive reversion toward consensus". The findings in this regard relate to a major ongoing debate in molecular evolution about the extent to which diverged homologs retain strong propensities for specific amino acids. Pollock et al. (PNAS, 109:E1352) and Shah et al. (PNAS, 112:E3226) have argued that pervasive epistasis means that the propensities for specific amino acids means will diverge rapidly among homologs, which will quickly come to favor different amino acids at homologous sites. On the other hand, Ashenberg et al. (PNAS, 110:21071), Risso et al. (Mol Biol Evol, 32:440), and Doud et al. (Mol Biol Evol, doi:10.1093/molbev/msv167) have argued that the propensities for specific amino acids are similar among homologs. How do your results bear on this issue?*

This is an interesting discussion and our results certainly suggest largely conserved amino acid preferences at the level of divergence observed between HIV-1 subtypes. This does not exclude that the preference at some sites have changed, but the majority is conserved. We have added a discussion of this issue.

*7) In the tree in Figure 1, near the yellow for patient 9 there is a black line that doesn't connect to any colored lines in the tree. What does this indicate?*

We apologize for the lack of explanation. This sequence is the HxB2 reference sequence. This is now explicitly stated in the caption.

*8) It is confusing that there is one Methods section called "diversity and divergence" and another called "divergence and diversity".*

Thank you very much for pointing out this oversight. We have now given these sections headings that more accurately reflect their content.

*9) The y-axis of Figure 5 should be labeled with the appropriate units, or this information should be added to the figure caption.*

We have added the units to the caption.

*10) You state: "Figure 6 indicates that diversity accumulates over a time frame of 2-4 years." However, since Figure 6 shows the rate of reversion compared to variability at 5-6 years, it is not clear how this demonstrates that diversity accumulates over a 2-4 year period.*

Thank you very much for spotting this. We meant to refer to Figure 5 which shows the time course of diversity.

*11) A*32 is overrepresented in this group; a much higher frequency than found in the broader Swedish population. Can the authors comment? Are these subjects in any way linked?*

We agree that the overrepresentation of A*32 is surprising. Dr Bratt has carefully checked if there could exist some relatedness or similar ethnic background of patients 8-11, but no such factors could be identified. Patients 4 and 7, who were sampled but excluded from the current analyses because of poor PCR performance or suspected superinfection, did not have the A*32 allele. Some reports (Geczy, Hum Immunol 2000;61:172-6; Lazaryan, Hum Immunol 2011;72:312-318) have indicated that A*32 is associated with long-term survival. Our patients had relatively slow disease progression and we cannot exclude that our inclusion criteria resulted in overrepresentation of certain HLA types among our patients. We have refrained from discussing this in the manuscript.